# In Silico and In Vivo Studies of a Tumor-Penetrating and Interfering Peptide with Antitumoral Effect on Xenograft Models of Breast Cancer

**DOI:** 10.3390/pharmaceutics15041180

**Published:** 2023-04-07

**Authors:** Gustavo H. Marin, Samuel Murail, Laura Andrini, Marcela Garcia, Severine Loisel, Pierre Tuffery, Angelita Rebollo

**Affiliations:** 1Department of Pharmacology/Histology and Embryology, FMC, National University of La Plata, CONICET, La Plata 1900, Argentina; 2BFA, Université Paris Cite, CNRS UMR 8251, Inserm U1133, 75013 Paris, France; 3Animalerie de l’UBO, UFR Médicine, 29238 Brest, France; 4Faculté de Pharmacie, UTCBS, Université Paris Cite, Inserm U1267, 75006 Paris, France

**Keywords:** breast cancer, tumor-penetrating peptide, protein–protein interaction in silico modeling

## Abstract

The combination of a tumor-penetrating peptide (TPP) with a peptide able to interfere with a given protein–protein interaction (IP) is a promising strategy with potential clinical application. Little is known about the impact of fusing a TPP with an IP, both in terms of internalization and functional effect. Here, we analyze these aspects in the context of breast cancer, targeting PP2A/SET interaction, using both in silico and in vivo approaches. Our results support the fact that state-of-the-art deep learning approaches developed for protein–peptide interaction modeling can reliably identify good candidate poses for the IP-TPP in interaction with the Neuropilin-1 receptor. The association of the IP with the TPP does not seem to affect the ability of the TPP to bind to Neuropilin-1. Molecular simulation results suggest that peptide IP-GG-LinTT1 in a cleaved form interacts with Neuropilin-1 in a more stable manner and has a more helical secondary structure than the cleaved IP-GG-iRGD. Surprisingly, in silico investigations also suggest that the non-cleaved TPPs can bind the Neuropilin-1 in a stable manner. The in vivo results using xenografts models show that both bifunctional peptides resulting from the combination of the IP and either LinTT1 or iRGD are effective against tumoral growth. The peptide iRGD-IP shows the highest stability to serum proteases degradation while having the same antitumoral effect as Lin TT1-IP, which is more sensitive to proteases degradation. Our results support the development of the TPP-IP strategy as therapeutic peptides against cancer.

## 1. Introduction

Breast cancer is still a major public health problem worldwide, with a need for new therapeutic options. This cancer is characterized by the uncontrolled growth of malignant cells in the mammary epithelial tissue. Surgical resection, chemotherapy, and hormonal or monoclonal antibodies are the most commonly used treatments. 

Protein phosphatase 2A (PP2A), one of the main serine/threonine phosphatases in mammalian cells, maintains the homeostasis by counteracting the signaling pathways of several kinases [1,2]. PP2A is genetically altered or functionally inactivated in several solid cancers and leukemias [3,4]. Inhibition of PP2A activity is critical to promote cell transformation, tumor progression and angiogenesis, indicating the tumor suppressor role of PP2A [5,6,7]. Several studies show that restoration of PP2A tumor suppressor activity antagonizes cancer development and progression [7]. Elucidating the mechanism of PP2A dysregulation in on diseases has led to increasing interest in the development of PP2A-targeted and selective therapeutics in cancer. 

SET is a multitask oncoprotein critically involved in the initiation and progression of cancer. SET is overexpressed in several types of cancers, including 60% of breast tumors. SET binds to PP2A and is a strong physiological inhibitor of the PP2A phosphatase activity [8,9,10]. SET plays a key role as an oncoprotein through regulation of its phosphorylation or over-expression [11].

Targeting and disrupting SET/PP2A interaction would result in a recovered PP2A activity and consequently reduced tumor growth. The novel compounds, FTY720, EMQA and the apolipoprotein mimetic peptide, targeting the PP2A/SET interface, inhibit tumor growth and overcome therapeutic resistance in many different malignant diseases. 

We have identified the binding site of PP2A to SET using PEP scan approach, characterized the peptide interfering with this interaction (IP) [12] and associated this sequence to four different tumor-penetrating peptides (TPP), generating four new tumor-penetrating and interfering peptides. We then validated their anti-tumoral effect on xenograft models of chronic lymphocytic leukemia models [13]. 

TPP are peptides that specifically recognize, bind and internalize into tumoral cells [14]. TPP are defined by the presence of the C-end rule (CendR) motif with the consensus sequence R/KXXR/K (X, any amino acid [15,16]. This position-dependent motif must be C-terminally exposed to allow for binding to receptor Neuropilin-1 (NRP-1) [17]. NRP-1 is overexpressed in the tumor vasculature and in a variety of malignant cells [18,19]. Several TPP have been described: RPARPAR, iRGD, TT1 and its linear version, Lin TT1 [17,20,21,22]. iRGD is recruited to the tumor vessels by interaction with integrins, and after this, it is cleaved by tumor-associated proteases to expose the CendR motif to allow interaction with NRP-1 [23]. TT1 and Lin TT1 bind first to p32, a protein expressed on the cell surface of tumoral cells [24,25,26], and are then cleaved by tumoral proteases in order to bind to NRP-1. 

Little is currently known about the impact of combining IP and TPP related to its binding to Neuropilin-1, and furthermore on the internalization of the IP-TPP and its biological effect. In this study, we combine in silico and in vivo experiments to ascertain the putative relationship between the binding of the TPP to NRP-1 and the antitumoral effect of the tumor-penetrating and interfering peptides iRGD-IP and Lin TT1-IP in xenograft models of breast cancer. 

## 2. Materials and Methods

### 2.1. Peptide Synthesis and Sequence

Peptides were synthesized in an automated multiple peptide synthesizer with solid phase procedure and standard Fmoc chemistry by GL Biochem (Shanghai, China). The purity and composition of the peptides were confirmed by reverse phase high-performance liquid chromatography (HPLC) and by mass spectrometry (MS). The sequences of the peptides used for in vitro and in vivo experiments are provided Table 1. Note that the IP part is constant, while only the TPP part differs.

Acronyms for the peptides used in the text are specified in column 1 and correspond to those of previous publications [27].

Aminohexanoic acid (Ahx) is used as a linker between the IP and the TPP since it is expected to afford some flexibility to the resulting bi-functional peptide [28]. The IP was associated to the N-ter of the TPP. The description of the identification of the IP was previously provided [12]. 

### 2.2. Peptide NRP-1 Complex Structure Prediction

To compute the structure of the Neuropilin-1 receptor in complex with IP-iRGD and IP-LinTT1 peptides, we decided to focus on the b1 domain of Neuropilin-1 which has been previously identified as [14]. The sequence Uniprot Code: P97333 was used with residues 275 to 424 to model domain b1 of Neuropilin-1. Since Alphafold only accepts genetically coded amino acids sequence, the 6-aminohexanoic acid (Ahx) linker could not be included in the sequence input. Moreover, to our knowledge, no data exist to optimize and validate the force-field parameters of the Ahx used as a linker in molecular dynamics simulations. We decided to replace it with the GG sequence, which we identified as a better mimic of the Ahx linker, at least when considering the geometric insertion. In fact, Ahx contains 6 rotatable bonds between the amine and the carboxyl group and is known to be flexible [28]. Di-glycine contains only 5 bonds, including one amide bond that is rather rigid, while tri-glycine would encompass 8 bonds, making it longer than Ahx. Since we are not attempting to derive any quantitative properties of the TPP-IP binding to NRP1, this approximation seems acceptable. A total of 8 peptides listed in Table 2 were docked on the b1 domain of Neuropilin-1. Note that for all sequences, the IP is unaffected.

Alphafold 2.2.4 [29] was used with the Colabfold 1.3.0 implementation [30]. Colabfold jobs were computed using the batch.py script provided, and the multiple sequence alignments (MSA) were computed using mmseqs2 [31] on the UniRef + Environmental sequence set and a pair-mode “unpaired + paired”, which generates separate MSA for receptor and peptide and pairs sequences from the same operon within the genome. The structures were computed using 9 recycles and the multimer-v2 weights. In order to increase complex structure sampling, the dropout option was activated enabling the stochastic part of the model, and 10 calculations were computed with a different random seed, resulting in 50 Neuropilin-1-peptide complex structures for each peptide sequence. Models were classified [32] by comparing the complex model structures to the crystal structure of Neuropilin-1 b1 domain in complex with the SARS-CoV-2 S1 C-end rule peptide [33] (peptide sequence NSPRRAR, PDB code: 7JJC). It is important to notice that the peptide sequence NSPRRAR meets the C-end rule, making the 7JJC PDB entry a reference for the interaction of the CendR motif with Neuropilin-1. The comparison was performed using the dockQ score [32]. High dockQ values correspond to models in better adequacy with the reference structure, 1 being the perfect agreement. Different quality categories (in correct, acceptable, medium and high qualities) are defined according to threshold values of 0.23, 0.49, and 0.8. For instance, models with a dockQ value between 0.49 and 0.8 are considered as medium quality. The predicted dockQ (pdockQ) scores were also computed using the confidence metrics provided by Aphafold software [34]. For pdockQ, only one threshold value of 0.23 separates incorrect mpdels from acceptable ones.

### 2.3. Molecular Dynamics Simulation

Structure preparation and protonation were performed using the pdbfixer module of the OpenMM package. The complexes were solvated in a truncated octahedron box with a padding of 1.0 nm and the TIP3P water model. The Amber14SB force field [35] was used to model the protein atoms. Sodium and chloride ions were added to counter the charge of the protein and to reach an ionic concentration of 150 mM. The simulations were calculated using the OpenMM 7.7 package [36]. Systems were minimized for 10.000 steps and equilibrated for 10 ns in the NPT ensemble using position restraints on all C_α_ atoms of 10.0 KJ·mol^−1^·nm^−2^. Temperature and pressure were equilibrated using a Monte Carlo barostat at a temperature of 300 K and a pressure of 1 atmosphere. We used an integration time step of 4 fs using heavy hydrogen that was assigned a mass of 3 atomic mass units. Productions were computed during 1 μs without any position restraints. For each simulated pose, four replicas were simulated. MD simulation analysis were conducted using the MDAnalysis python library [37].

### 2.4. Mice

Adult male C3H/S strain mice of 8 weeks were used for the experiments. Mice were weighed daily until the end of the experiment and sacrificed on day 41. Mice were sacrificed on day 41. The animals were kept under standard conditions: water and food available ad libitum, and alternation of light and dark periods of 12 h each.

### 2.5. Breast Cancer Xenograft Models and Treatment

After an appropriate synchronization period, the C3H/S adenocarcinoma cell line was subcutaneously grafted into the flank of the animal. This is a breast carcinoma spontaneously originated in 16 months old C3H/S mouse. Tumor-bearing animals were divided into three groups: (i) a control group, intraperitoneally injected with saline buffer; (ii) a group injected with LinTT1-IP peptide at 5 mg/kg; (iii) a group injected with the peptide iRGD-IP at 5 mg/kg. The treatment started 3 days post adenocarcinoma grafting and continued for 11 days. When the tumors were around 0.2 mm in size, tumor measurement was performed daily until the end of the experiment. 

### 2.6. Analysis of Peptide Stability in Human Serum 

The peptide TT1-IP was incubated at 37 °C in 250 mL of human serum (Gibco, Illikirch, France) for different periods of time. Samples were collected and peptide degradation stopped by freezing at −20 °C. The peptide was extracted from samples using the Proteo Miner Protein Enrichment System (Bio-Rad, Marnes-la-Coquette, France). The percentage of intact peptide was estimated by mass spectrometry (MS) using MALDI-TOFF with the protocol previously described [38] (Bruker Autoflex II, Palaiseau, France) according to the manufacturer’s instructions. Measurements were performed in triplicate. MS data were analysed using the software Cliprot tools, Flex analysis, Bruker. 

### 2.7. Histological Samples Staining

Samples were processed for staining with hematoxylin and eosin according to the following steps: Dehydration by passages in alcohols of increasing strengths, 70°, 96° (two passages), 100° (two passages). Xylol (two passages) was used as an intermediate and clarifying liquid. The time required to remain in each of the liquids depended on the thickness and size of the sample. The impregnation was performed in high melting point paraffin (56 °C to 58 °C), kept in an oven at constant temperature. Cassettes were used for the final embedding in paraffin. Several 5 µm sections were produced with a Minott microtome. For deparaffinization, xylol and alcohols of decreasing strengths were used, ending with distilled water. The coloration was performed using Harris hematoxylin and yellow eosin in aqueous solution. The stock hematoxylin solution was diluted in distilled water (1:4) and eosin was used at 1% in distilled water. Staining was performed as follows: deparaffinized and hydrated sections were placed in Harris’ hematoxylin solution for 3 to 5 min. They were then immersed in running water to produce the dye toning. The sections were then placed in the eosin solution for 1 min. Finally, dehydration, rinsing and mounting were carried out.

### 2.8. Statistical Analysis

The data were analysed using ANOVA and Student–Newman–Keuls Multiple Comparison test. 

### 2.9. Ethical Considerations

Conditions concerning animal management fully respected the policy and mandates of the Guide for the care and use of laboratory animal research of the National Research Council.

## 3. Results

### 3.1. Neuropilin1–Peptide Interaction Models

Each of the eight peptides of Table 2 was modelled in a complex with the b1 domain of mouse Neuropilin1. The cleaved peptides correspond to the sequence generated upon cleavage by tumoral proteases. As shown in Figure 1A, the dockQ score is on average higher for the cleaved form, suggesting some slight structural impacts of the cleavage on the way the peptide binds to Neuropilin-1. It has been shown that in order to target Neuropilin-1, these peptides must be cleaved to bind to Neuropilin-1 [16], but from the docking experiments, little difference could be observed. When considering the impact of the GG linker, one observes a slight improvement for best poses obtained using the iRGD peptide, whereas there seems to be almost no difference for LinTT1. This suggests, however, some positive impact of the linker, or at least no negative impact on the ability of the TPP to bind the Neuropilin-1. 

The possibility to identify the best poses from prediction is depicted in Figure 1B,C. Similar results were obtained for the other systems involving the peptides in Table 2. For both peptides, IP-GG-iRGD cleaved and IP-GG-nTT1cl, we can see a good correlation between dockQ and pdockQ scores (*p*-values < 10^−8^) (Figure 1B,C, respectively). The best dockQ scores reach values around 0.7; this value is considered as “Medium Quality” and is close to the “High Quality” threshold of 0.8. However, considering the pdockQ, the 0.23 threshold of “Acceptable Quality” could not be reached, as best values could only reach 0.2. For the IP-GG-LinTT1 cleaved, the best dockQ and pdockQ scores were obtained for the same pose (0.708 and 0.193, respectively). Overall, the combination of TPP and PI does not appear to disrupt the binding of TPP to NRP1and; although, obtained for only a small number of systems, these results suggest that the predicted dockQ score can be effective in identifying good candidate poses. 

Of note, in all best models in terms of dockQ or pdockQ score, the peptide always displays a similar orientation of the last charge C-ter residues (R26 and K25 for I-GG-iRGD and IP-GG-LinTT1 peptides, respectively), with the positively charged residue side chain pointing toward the center of Neuropilin-1 and in close proximity to D320 of Neuropilin-1 (see Appendix A).

### 3.2. Molecular Dynamics Simulation

Good or medium quality poses do not imply the stability of the binding. This was investigated using molecular dynamics (MD) simulations. For the IP-GG-LinTT1-cl, the best dockQ and pdockQ model was selected to undergo simulation. As for IP-GG-iRGD-cl, we decided to simulate the best dockQ and best pdockQ score models. the two models had dockQ and pdockQ values of 0.697, 0.103, and 0.632, 0.199, respectively. For clarity, the best dockQ model will be called IP-GG-iRGD-cld, while the best pdockQ model will be called IP-GG-iRGD-clp. The uncleaved peptides were also simulated. For the IP-GG-LinTT1 case, we decided to simulate the best pdockQ model (pdockQ of 0.239 just above the pdockQ threshold and a dockQ of 0.634). For the IP-GG-iRGD case, we decided to simulate the best dockQ structure (dockq and p dockQ of 0.511 and 0.0095, respectively). For all simulated models, it should be noted that the peptide with two positively charged residues sidechains (R22 K25 for iRGD and R23 R26 for LinTT1, Figure 2) are pointing toward D320, the same residue that form a salt bridge with the peptide’s arginine in PDB structure 7JJC. 

The stability of the protein–peptide complexes was determined by monitoring the root mean square deviation (RMSD) to the initial structures of the protein and peptide. For the five models, IP-GG-LinTT-cl, IP-GG-iRGD-cld, IP-GG-iRGD-clp, IP-GG-LinTT1 and IP-GG-iRGD, the Neuropilin-1 protein was particularly stable with a low backbone RMSD of 1.25 ± 0.09 Å, 1.44 ± 0.14 Å, 1.52 ± 0.20 Å, 1.28 ± 0.16 Å and 1.31 A ± 0.20 Å, respectively. The peptide stability displays higher drift to the initial model, with a backbone RMSD of 13.94 ± 4.31 Å, 25.19 ± 7.21 Å, 13.79 ± 5.58 Å, and 16.35 ± 7.97 Å and 19.73 A ± 9.89 Å, respectively. Detailed RMSD plots for protein and peptide are shown for IP-GG-LinTT1-cl and IP-GG-iRGD-clp simulations in Appendix A. For IP-GG-iRGD-cld, the very high RMSDs could be explained by the fact that for two out of the four simulations, the peptide drifted away from the binding site at the beginning of the simulation (at 12 and 35 ns) and did not return to it. We thus decided to focus our analysis only on IP-GG-LinTT1-cl, IP-GG-iRGD-clp, IP-GG-LinTT1 and IP-GG-iRGD simulations. Of note, for one of the four simulations of the uncleaved form IP-GG-iRGD, the peptide left the binding site at the end of the simulation (after 800 ns), which explains the high RMSD value of close to 20 Å.

When examined in detail, the RMSD differs greatly depending on the position of the residues (Figure 2A and Figure 3A), the C-ter residues which contain the Neuropilin-1 targeting sequence (iRGD and LinTT1) had lower RMSD, reaching values bellow 5 Å for the four simulations of IP-GG-LinTT1-cl and for two out of four simulations of IP-GG-iRGD-clp. Interestingly, for IP-GG-LinTT1-cl, the most stable residue was the last arginine (R26), while for IP-GG-iRGD-clp, the most stable position was the penultimate charged residue (R22). Uncleaved peptides exhibit similar behavior, with the R/KXXR/K motif corresponding to the most stable residues. However, the cleaved residues (last three residues) show higher drift.

Altogether, the stability of the C-ter peptide is consistent with the role of R/KXXR/K motif (iRGD and LinTT1) targeting sequence of Neuropilin1 and can be considered as the anchoring sequence; by contrast, the N-ter part displayed higher structural drift, reaching RMSD of up to 35 Å, and its binding could be considered as less than or not specific to Neuropilin-1. In the available structures of the PDB of Neuropilin1 in complex with a peptide, only the last two or three residues of the peptide could be reconstructed in the density map, indicating high structural drift of the N-ter part of the peptide.

We determined the secondary structure of the peptides by calculating the fraction of residues in helical, beta-sheet, and coiled conformations (Figure 2B and Figure 3B). For the four peptides, the IP peptide part was mostly structured as an alpha helix, whereas the iRGD and LinTT1 parts were mostly unstructured. IP-GG-LinTT1-cl was determined to have a higher fraction of helical residues compared to IP-GG-iRGD-clp. For the uncleaved form of IP-GG-iRGD, a non-negligible fraction of helical residues could also be observed in the C-ter part.

We also analyzed that salt bridges between Neuropilin1 and peptides were more frequent with the peptide C-ter (Figure 2C and Figure 3C, mode details in Appendix A). For IP-GG-iRGD-clp, the R26-D320 was present in 70% of MD frames, with a similarity to the PDB structure 7JJC, as the R23-D320 was also present in 47% of MD frames. In the uncleaved form, the R26-D320 salt bridge frequency drops to 0.5%. Neuropilin1 D320 residue was also involved in 47% of the frames in a salt bridge with R23 (50% in the non-cleaved form). This suggests that the presence of salt bridges contributes to the stability of the interaction.

Interestingly, some bridges not involving the TPP residues, such as E15 of IP-GG-iRGD-cl or R12 of IP-GG-LinTT1-cl and IP-GG-LinTT1, are also observed (Figure 2D and Figure 3D). It is unknown whether such interactions could interfere with peptide internalization and result in some IP modulation of the phenomenon.

Overall, our results suggest that IP-GG-LinTT1-cl is more stable and has a more helical secondary structure than IP-GG-iRGD-clp. They also suggest that the presence of residues anchored with salt bridges can strongly influence the stability and conformation of protein/TPP-IP complexes, while the N-ter residues can explore a large structural space (main conformations of IP-GG-LinTT1-cl and IP-GG-iRGD-clp are shown in Appendix A, respectively). The uncleaved peptides display a slightly higher RMSD compared to the cleaved form; however, we miss the really conclusive sampling on that point. More divergence is however observed for IP-GG-LinTT1 when compared to that of IP-GG-iRGD.

### 3.3. Antitumoral Effect of Lin TT1 and iRGD Interfering Peptides on Breast Cancer Xenograft Models

Given that both peptides show similar binding stability to the receptor, we decided to test whether they had a similar antitumoral effect in vivo. The antitumoral effect of the peptides Lin TT1-IP and iRGD-IP was evaluated in a xenograft mouse model of breast cancer generated using the cell line C3H/S. The mice (eight per group) were treated with the corresponding peptide using a dose of 5 mg/kg for 11 days or with saline buffer (control group). As shown in Figure 4, treatment of the breast cancer-bearing mice with Lin TT1-IP and iRGD-IP peptides had a therapeutic effect with a diminution in the size of the tumor compared to control group, starting at J32 after injection. In addition, both peptides iRGD-IP and Lin TT1-IP show comparable antitumoral activity. The differences between treated and control mice were statistically significant. 

Breast cancer xenograft models were intraperitoneally (IP) injected with the peptides at 5 mg/kg starting three days after the graft. Growth of the tumor was monitored over time. Statistical analysis and comparisons were made using ANOVA (*n* = eight animals per group). * *p* < 0.01 treatment versus vehicle control group.

To assess the potential toxicity of the peptides, we compared the dynamic body weight of the mice in different treatment groups. As shown in Figure 5, the mice started to lose weight between days 30 and 42. No significant body weight change was observed in the treated mice compared to the control group, suggesting that the weight loss occurred due to the disease development and not the peptide toxicity. 

Weight of the breast cancer xenograft mice treated with the peptides or saline solution was monitored every two days. Each value is represented as the mean ± standard deviation (SD) (*n* = 8).

### 3.4. Stability of iRGD-IP and Lin TT1-IP Peptides in Serum

Proteolytic degradation of peptide-based drugs is considered as a major problem for the development of peptides as medicament, often limiting their therapeutic applications. We analyzed the stability of iRGD-IP and Lin TT1-IP peptides upon incubation at 37 °C in human serum. Figure 6 shows that around 80% of the total initial amount of Lin TT1-IP peptide is recovered after 3 h of incubation, 44% at 6 h and around 30% after 24 h of incubation. However, the peptide iRGD-IP looks more resistant to proteases degradation, recovering 91% upon 1 h of incubation, 77% upon 6 h and 62% upon 24 h of incubation in human serum at 37 °C. 

iRGD-IP and Lin TT1-IP were incubated in human serum at 37 °C for different periods of time. The integrity of the peptide was analyzed by mass spectrometry (MS). The percentage of the recovered peptide over time is presented in the figure. Every measurement was performed in triplicate. Each value is represented as the mean ± standard deviation (SD). Similar results were obtained in two independent experiments. 

### 3.5. TPP-IP Peptides Induce Apoptosis of Tumoral Graft 

We analyzed the histological status of the control tumor and the treated tumor (Figure 7). In the control non-treated tumor, we observed a solid, encapsulated neoplasm with medium to large cells, vesicular nuclei with prominent nucleoli and little basophilic cytoplasm. Mitosis and apoptotic forms were observed. In tumors treated with peptide iRGD-IP, we observed an atypical cell proliferation arranged in nests and cords, composed of medium-sized, polyhedral cells with intense eosinophilic cytoplasm and marked anisokaryosis, with the presence of moderate nuclear pleomorphism. In addition, numerous mitotic figures and abundant apoptotic bodies were identified. The neoplasm was accompanied by extensive areas of necrosis of predominantly central distribution, together with a mixed inflammatory infiltrate, which was mainly distributed in the periphery.

Tumor from control or iRGD-IP-treated mice were stained with hematoxylin/eosin and analyzed by microscopy. The scale bar is shown.

## 4. Discussion

Protein–protein interactions are emerging as a new molecular targeting strategy for many therapeutic areas including cancer. Tumor-penetrating peptides have emerged as a new class of vectors that allow the delivery of molecules to specific tumor cells. 

An exciting target approach for cancer therapy is the use of bi-functional peptides with tumoral penetration and protein–protein interfering properties. We have previously published the use of bi-functional peptides associated to a cell penetrating peptide or to a tumor-penetrating peptide. These peptides were validated in xenograft models of chronic lymphocytic leukemia (CLL) (TPP shuttle) or breast cancer (CPP shuttle) [27].

PP2A has been characterized as a tumor suppressor and inhibitions of PP2A activity increases tumorigenesis, suggesting that modulation of PP2A activity can be beneficial for cancer treatment [39,40]. It has been shown that pharmacological restoration of PP2A activity is able to kill tumor cells [41,42]. Several inhibitors of PP2A activity have been described, among them the oncoprotein SET. This protein contributes to tumorigenesis by forming an inhibitory complex with PP2A [8,43,44]. Several efforts to restore PP2A activity are focused on interfering PP2A/SET interaction [11,45]. Therefore, PP2A/SET interaction is a promising protein–protein interaction to be modulated as a therapeutic target. 

OP449, a chimeric peptide corresponding to a CPP coupled to a sequence of apolipoprotein E, was reported to bind SET as an apoE-mimetic preventing PP2A/SET interaction [43]. In our case, we have used a specific TPP associated to an interfering peptide specifically designed to target PP2A/SET interaction [12]. By adopting this targeting strategy, we might not affect other SET partners/functionalities. We have previously preclinically validated a tumor-penetrating and interfering peptide that blocks PP2/SET interaction against CLL [13]. Given that this interaction is also dysregulated in breast cancer, we decided to test the antitumoral effect of two TPP-IP, iRGD-IP and Lin TT1-IP in xenograft models of breast cancer. Lin TT1-IP and iRGD-IP are proteolytically cleaved by tumor enzymes after binding to p32 and integrins, respectively. The cleavage activates the CendR motif to be able to interact with NRP-1 to trigger the internalization of the peptide/cargo into tumor cells. 

From our in silico analyses, an important outcome was that we could observe only slight differences in the binding of the non-cleaved and cleaved peptides for an aggregated simulation time of 4 μs for each complex. Since we cannot assure that the sampling is sufficient for conclusion, the employment of advanced sampling techniques such as replica exchange molecular dynamics or simulated tempering might be useful to better characterize those slight differences.

However, both cleaved and non-cleaved peptides seemed able to bind the NRP-1 R/KXXR/K experimental binding site in a stable manner which questions the role of the cleavage. One possible explanation could be in terms of the kinetics of the cleavage, which could occur faster than the binding to NRP-1, not being mandatory. Another direction is, of course, the impact of the slight differences observed on the binding of the two forms on the molecular behavior of NRP-1 and its consequences on internalization.

Another observation from the in silico experiments concerns the role of the linker, which seems to ease the binding of the IP-TPP to NRP-1. Our results suggest that further investigations on its nature could be of interest, particularly since the use of a di-glycine as linker for the simulations is only an approximation of the Ahx used for in vitro and in vivo experiments. Despite this, as can be seen from Appendix A, we indeed observed a large diversity of conformations and relative orientations of the IP with respect to the TPP. The extent to which such flexibility could contribute to the internalization remains to be further investigated.

Finally, a final observation from the in silico analyzes is the rather similar behaviors of IP-iRGD and IP-LinTT1 related peptides. Here, again, we cannot assure that simulation times are sufficient for conclusion, but the binding of IP-iRGD and IP-LinTT1 peptides seems to be clearly mediated by salt bridges. It would be interesting to investigate further to what extend the strength of the binding is or is not related to peptide internalization and to peptide biological activity.

Our experimental results show that both tumoral-addressed peptides iRGD-IP and LinTT1-IP have an antitumoral effect on xenograft models of breast cancer, suggesting a potential clinical application. Compared to other tumoral targeting therapies such as antibodies, peptides offer several advantages, including small size, low toxicity, low manufacturing cost and low immunogenicity. iRGD-IP shows the highest stability to serum protease degradation having the same antitumoral effect as Lin TT1-IP, which appears more sensitive to protease degradation. Overall, combining lessons from our in silico experiments with those from our in vitro and in vivo experiments opens the door to further investigations to better understand the TPP-IP strategy and to rationally design more effective bi-functional peptides.

## Figures and Tables

**Figure 1 pharmaceutics-15-01180-f001:**
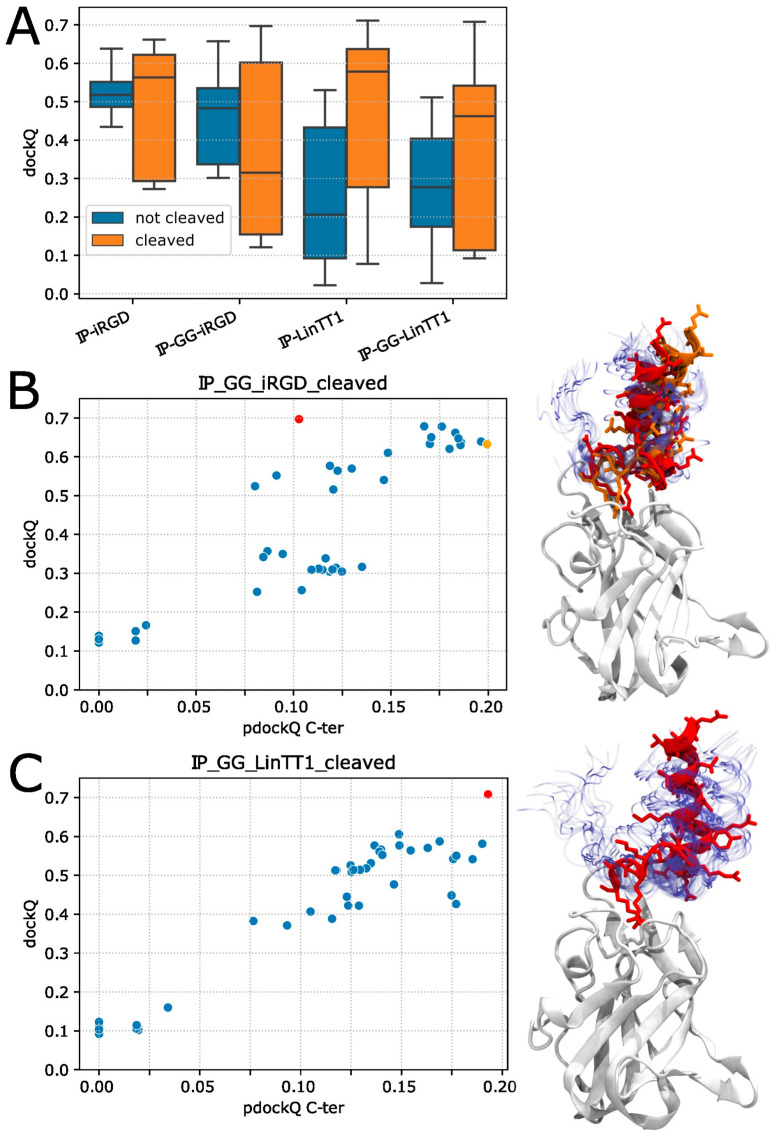
Quality measurement of Alphafold models. (**A**) Boxplot of the DockQ score, the four peptides IP-iRGD, IP-GG-iRGD, IP-LinTT1 and IP-GG-LinTT1 (50 alphafold models each) were studied in their uncleaved and cleaved forms and displayed as blue and orange boxplots, respectively. Scatterplot of DockQ versus pdockQ scores for the 50 models of IP-GG-iRGD-cl (**B**) and IP-GG-LinTT1-cl (**C**), the red and orange dots show the best models according to their dockQ and pdockQ scores. The left panels display the 50 superimposed alphafold models, Neuropilin-1 is shown as a white cartoon, as the 50 peptide models are shown as blue transparent ribbons, the best peptides are shown in colored cartoons according to scatterplot colors (orange and red); for the same models, the peptide atoms are shown as colored sticks. Larger images of the 3D structures are available as Appendix A.

**Figure 2 pharmaceutics-15-01180-f002:**
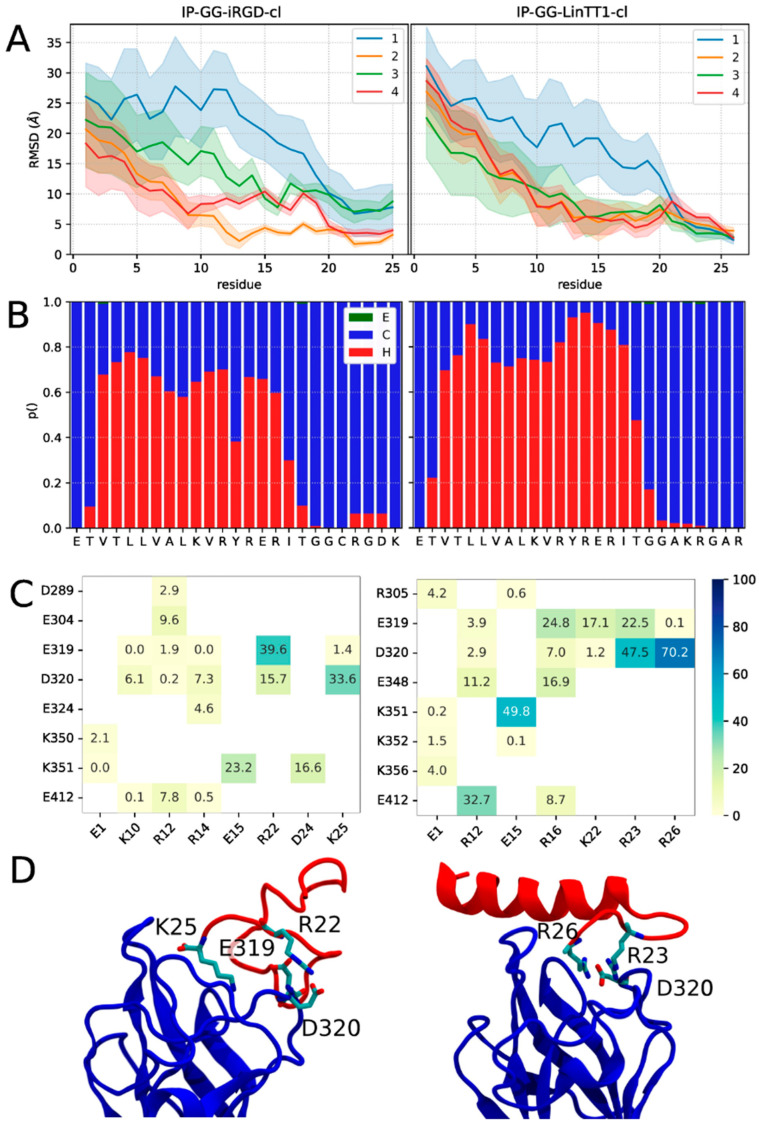
Cleaved Peptide stability secondary structure and interaction with Neuropilin1. (**A**): Peptide RMSD per residue of IR-GG-iRGD-cl and IR-GG-LinTT1-cl, colored by simulation replica; error bar represents the standard deviation of RMSD. (**B**) Secondary structure percentage of IR-GG-iRGD-cl and IR-GG-LinTT1-cl (E, C and H stand for extended strain, coil and helix secondary structure, respectively). (**C**) Frequency of salt bridge interaction between Neuropilin1 (*y*-axis) and peptides (*x*-axis). Only salt bridges with frequencies higher than 0.01 are shown. (**D**) Representative structure of the biggest cluster. All MD simulation frames were clustered based on C-ter residue backbone structure.

**Figure 3 pharmaceutics-15-01180-f003:**
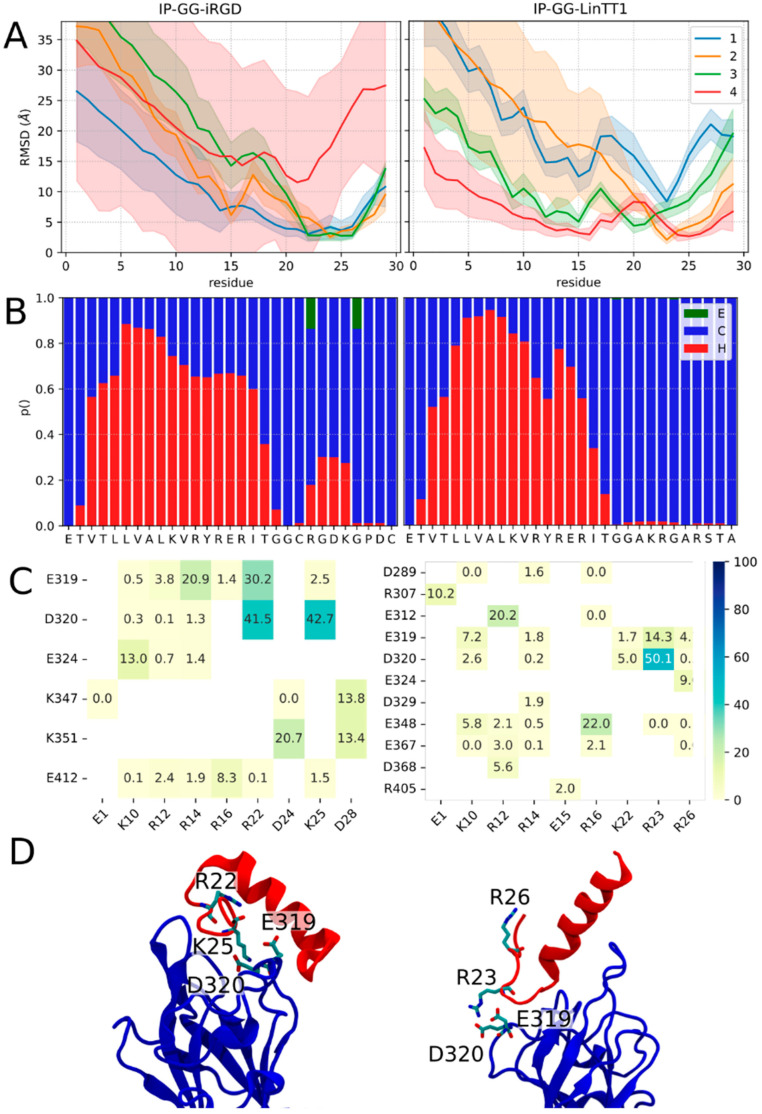
Non-cleaved Peptide stability secondary structure and interaction with Neuropilin1. The analyses are identical to those in Figure 2, for IR-GG-iRGD and IR-GG-LinTT1. (**A**) Upper panel: peptide RMSD per residue, colored by simulation replica; error bar represents the standard deviation of RMSD. (**B**) Secondary structure percentage of IR-GG-iRGD and IR-GG-LinTT1. (**C**) Frequency of salt bridge interaction between Neuropilin1 (y-axis) and peptides (x-axis). Only salt bridges with a frequency higher than 0.01 are shown. (**D**) Representative structure of the biggest cluster. All MD simulation frames were clustered based on the C-ter residues backbone structure.

**Figure 4 pharmaceutics-15-01180-f004:**
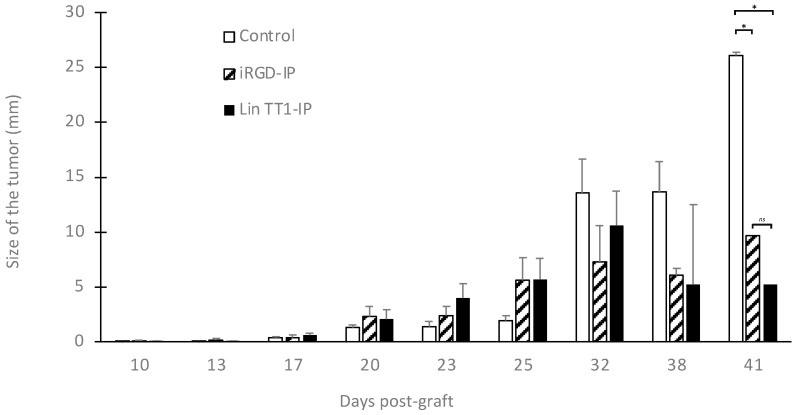
Antitumoral effect of iRGD-IP and Lin TT1-IP peptides. * *p* < 0.01 treatment versus vehicle control group. ns: no significant.

**Figure 5 pharmaceutics-15-01180-f005:**
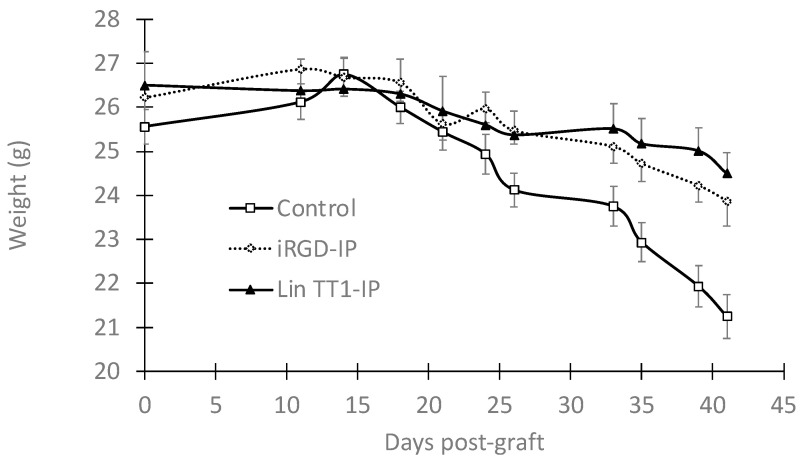
Effect of the peptide on the body weight.

**Figure 6 pharmaceutics-15-01180-f006:**
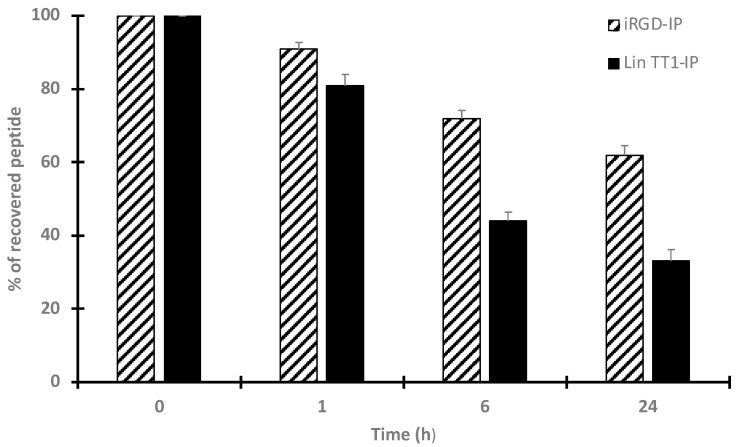
Stability of the peptides in human serum.

**Figure 7 pharmaceutics-15-01180-f007:**
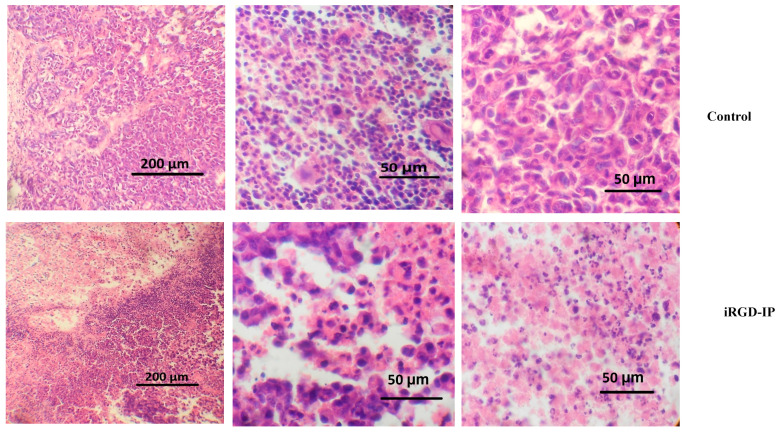
Histological analysis of TPP-IP treated breast cancer xenograft models.

**Table 1 pharmaceutics-15-01180-t001:** Sequences considered for experimental studies.

iRGD-IP	ETVTLLVALKVRYRERIT-Ahx-**C**RGDKGPD**C**-CONH_2_ (**C-C** disulfide bond)
LinTT1-IP	ETVTLLVALKVRYRERIT-Ahx-AKRGARSTA-CONH_2_

**Table 2 pharmaceutics-15-01180-t002:** Sequences considered for the in silico modeling and simulations. Acronyms for each peptide are specified in column 1 and, by contrast with those of the sequences used for in vitro and in vivo experiments, follow the sequence order. For instance, IP-iRGD-cl is hereafter used for a peptide combining the IP (ETVTLLVALKVRYRERIT) followed by the cleaved TPP (CRGDK) without any linker between the two. Similarly, IP-GG-LinTT1 stands for IP (ETVTLLVALKVRYRERIT) followed by a GG linker, and then by an uncleaved TPP (AKRGARSTA).

IP-iRGD (no linker, full iRGD)	ETVTLLVALKVRYRERITCRGDKGPDC
IP-GG-iRGD (GG linker, full iRGD)	ETVTLLVALKVRYRERIT**GG**CRGDKGPDC
IP-iRGD-cl (no linker, cleaved iRGD)	ETVTLLVALKVRYRERITCRGDK
IP-GG-iRGD-cl (GG linker, cleaved iRGD)	ETVTLLVALKVRYRERIT**GG**CRGDK
IP-LinTT1 (no linker, full LinTT1)	ETVTLLVALKVRYRERITAKRGARSTA
IP-GG-LinTT1(GG linker, full LinTT1)	ETVTLLVALKVRYRERIT**GG**AKRGARSTA
IP-LinTT1-cl (no linker, cleaved LinTT1)	ETVTLLVALKVRYRERITAKRGAR
IP-GG-LinTT1-cl (GG linker, cleaved LInTT1)	ETVTLLVALKVRYRERIT**GG**AKRGAR

## Data Availability

Not applicable.

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
