# Peer review of "In Silico and In Vivo Studies of a Tumor-Penetrating and Interfering Peptide with Antitumoral Effect on Xenograft Models of Breast Cancer"

_pharmaceutics, 2023, doi:10.3390/pharmaceutics15041180_

Round 1
Reviewer 1 Report
Angelita Rebollo and colleagues submitted a paper focused on a potential and rather innovative anticancer target, consisting in the SET/PP2A assembly. The paper is overall well written and interesting, but some improvements are needed before acceptance.
A reader which is confident with the SET/PP2A field could notice that literature search was not performed adequately. In fact, the authors should consider important papers in the field concerning the pharmacological potential of PP2a (O’Connor, Int J Biochem Cell Biol 2018 10.1016/j.biocel.2017.10.008), the molecular mechanisms of small molecules targeting SET/PP2A assembly (De Palma, FASEB 2019 10.1096/fj.201802264R) and experimental/preclinical studies of compounds with anticancer activity targeting this pathway (Pagano, Leukemia 2019 10.1038/s41375-018-0288-5; Zonta, Blood 2015 10.1182/blood-2014-12-619155; Tibaldi, Hematologica 2017 10.3324/haematol.2016.155747). In general, literature is outdated and must be improved.
A drawback affects the computational studies part, since, as stated by the authors, 6-aminohexanoic acid (Ahx) linker was not inserted in the 3D structure. This is an approximation that could affect the result. As for the results of docking, Figure 1 must be modified as the quality is too low to interpret data. Also data from Figure 2 and 3 cannot be read appropriately.
Overall, the quality of figures is very low and resolution must be improved.
Minor points:
Figure 3: the caption lacks of the explanation for the panels.
Page 6, line 235: please remove the yellow highlighting
Page 8, line 296: the authors state that “The difference between treated and control mice were statistically significative” but no statistical analysis is reported in the figure.
Academic Editor Notes
The authors have revised the manuscript and have satisfactorily addressed most of the issues raised by the four reviewers. All modifications have been made to deal with all the suggested minor issues. The clarity/quality of the Figures has been much improved and as too has the English throughout the text. The discussion has been improved and some brief concluding remarks given. There are a few issues that remain to be dealt with, specifically:
The title for the article should be changed as "In silico and in vivo studies of a tumor-penetrating ..."
There are numerous errors in the abstract to be corrected:
line 3, "few little" -> "little"
line 21, "of for" -> "for", line
line 24, "than of" -> "than"
line 26, "can are able to" -> "can"
line 29, "the having" -> "having"
line 30, "that which" -> "which"
On page 11, "statistically significative" should read "statistically significant", and the authors should give the confidence level in brackets, e.g., "... statistically significant (p < 0.001)." Also, the legend to Figure 4 should include the meaning of the asterisks (i.e., the confidence level of the marked differences should be specified here too.)
The legends to Figures 5 and 6 should state whether the error bars shown are standard deviations or standard errors, and the numbers of replicates should be specified.
Once these additional changes have been made, I consider that the revised manuscript can be accepted for publication.
Obj: pharmaceutics-2197144
Dear Pharmaceutics Editor,
We would like to thank you very much for reconsidering our manuscript entitled In silico and in vivo studies of a tumor penetrating and interfering peptide with antitumoral effect on xenograft models of breast cancer by Marin et al., and for your positive feedback on the article.
We apologize for the errors in the abstract but it seems that in the version that we sent by email to the editorial office, the words to be deleted were marked as crossed out words and the words modified were labeled in red. In the version that is available from the web site, all these marks have disappeared and the words have remained, generating some confusion. We have now fixed this definitely. The modifications are labelled in red.
In the new version of the manuscript, the following modifications have been made, according to your remarks:
- The title of the manuscript has been modified
- The statistical significance on figure 4 has been added on the legend of the figure.
- For figures 5 and 6, the information concerning the error bars has been added.
We have added on the web site the file with the supplementary information.
Reviewer 2 Report
The main problem with the manuscript is the poor quality of the images in Figures 1, 2 and 3, which makes the authors' description of the results of the work incomprehensible. The article needs to improve the discussion of the results, moreover the manuscript does not contain any conclusions.
Academic Editor Notes
The authors have revised the manuscript and have satisfactorily addressed most of the issues raised by the four reviewers. All modifications have been made to deal with all the suggested minor issues. The clarity/quality of the Figures has been much improved and as too has the English throughout the text. The discussion has been improved and some brief concluding remarks given. There are a few issues that remain to be dealt with, specifically:
The title for the article should be changed as "In silico and in vivo studies of a tumor-penetrating ..."
There are numerous errors in the abstract to be corrected:
line 3, "few little" -> "little"
line 21, "of for" -> "for", line
line 24, "than of" -> "than"
line 26, "can are able to" -> "can"
line 29, "the having" -> "having"
line 30, "that which" -> "which"
On page 11, "statistically significative" should read "statistically significant", and the authors should give the confidence level in brackets, e.g., "... statistically significant (p < 0.001)." Also, the legend to Figure 4 should include the meaning of the asterisks (i.e., the confidence level of the marked differences should be specified here too.)
The legends to Figures 5 and 6 should state whether the error bars shown are standard deviations or standard errors, and the numbers of replicates should be specified.
Once these additional changes have been made, I consider that the revised manuscript can be accepted for publication.
Obj: pharmaceutics-2197144
Dear Pharmaceutics Editor,
We would like to thank you very much for reconsidering our manuscript entitled In silico and in vivo studies of a tumor penetrating and interfering peptide with antitumoral effect on xenograft models of breast cancer by Marin et al., and for your positive feedback on the article.
We apologize for the errors in the abstract but it seems that in the version that we sent by email to the editorial office, the words to be deleted were marked as crossed out words and the words modified were labeled in red. In the version that is available from the web site, all these marks have disappeared and the words have remained, generating some confusion. We have now fixed this definitely. The modifications are labelled in red.
In the new version of the manuscript, the following modifications have been made, according to your remarks:
- The title of the manuscript has been modified
- The statistical significance on figure 4 has been added on the legend of the figure.
- For figures 5 and 6, the information concerning the error bars has been added.
We have added on the web site the file with the supplementary information.
Reviewer 3 Report
The authors present in-silico analysis of the binding of NRP-1 receptor to synthetic bifunctional peptides that combine a tumor penetrating peptide fragment (TPP) with another fragment that interferes with protein-protein interactions (IP) commonly seen in cancerous cells such as that between PP2A and SET. They also present in-vivo support for the antitumoral properties of the optimal sequences as well as their general serum stability. I find the conclusions logical and well supported by the data presented.
The authors should label the two tables with sequences in pages 2 and 3 respectively, for easier cross reference.
I have some questions and concerns:
1. Lines 81-84: It is unclear how the authors obtained the IP sequences that are grafted to the iRGD and LinTT1 sequences with or without the GG linker. Are these derived from their previously published analysis of functional peptides that disrupt SET-PP2A interaction, in Ref. 8? If so, please add a sentence explaining this clearly, otherwise the source of the IP sequences are confusing. Also, please name the two IP sequences differently. Calling both of them “IP” may lead to ambiguity for the reader.
2. Line 93-94: How do the authors decide the cleavage sites on the IP part of the sequence? If the fragments cleaved away (GPDC for IP-iRGD and STA for IP-LinTT1) are supported by data presented in Refs. 8 and 9, please explain the reason clearly.
For the GPDC deletion in IP-iRGD, the peptide sequences IP-iRGD-cleaved and IP-GG-iRGD-cleaved now have a free cysteine in each of them, which may lead to oxidative stress during synthesis and is commonly avoided in peptide sequence design. Can the authors comment on why they retain such an unpaired cysteine?
In the naming of the sequences, please maintain consistency with Lines 81-84, by first naming the TPP part then the IP part, i.e. iRGD-IP instead of vice versa.
Can the authors comment on the N-C terminal direction of the grafting? In other words, does it matter if the N-ter is the TPP (iRGD or LinTT1) and the C-ter is the IP (like in the paper) or vice versa? Since one of the requirements of the TPP structure is to maintain a C-terminal exposed basic residue, isn’t that achieved more easily by grafting the N-terminal of the TPP to the IP?
4. Line 105: “Important to notice that this peptide meets the CendRule”: Are the authors talking about the fragment NSPRRAR from the PDB ID: 7JJC or the peptide sequences designed in this study?
In fact, was it verified from the AlphaFold predicted structures (page5, Section 3.1) if the predicted peptide structures indeed have an exposed C-termini in the TPP part?
Can the authors also quantify the relative change in CendR motif exposure in the NRP1-peptide co-complexes over the course of the MD simulations?
6. In the molecular dynamics simulations (Section 3.2, page 5), for peptides that did not leave the binding site but still had large drifts in RMSD, can the authors quantify change in (peptide-NRP1) contact map or shape of the peptide epitope on NPR-1 over the course of the MD simulation? Since the starting models are not really physics-based protein-protein docks and the experimental structure of the co-complex is unknown, such an investigation may help get a coarser but more reliable picture of the peptide binding stability even for the high RMSD cases.
7. How did the authors select the linker length, i.e a single Ahx residue, or equivalently a (GG)n motif with n=1? The authors do talk about exploring the role of the linker in Lines 377-378, and linker length optimization is an entirely different project by itself, but I’m still curious about the initial choice of the linker length and would like the authors to provide an explanation in the text.
8. Finally, I have an observation that I would like the authors to comment on. They may or may not include that comment in the text, this is mostly for my own understanding:
In lines 65-67 the authors explain how TPPs first associate with integrins to gain entry into tumor vessels following a proteolytic cleavage to expose the CendR motif. In that case, I wonder if are there avenues to engineer proteolytic sequence motifs like RR closer to the TPP C-terminus (perhaps adjacent to the R/K-XX-R/K CendR motif) to promote easier protease action to expose the CendR, but also in a way that avoids drastic proteolytic degradation in the serum and maintains proper serum stability as observed in Fig. 6?
Academic Editor Notes
The authors have revised the manuscript and have satisfactorily addressed most of the issues raised by the four reviewers. All modifications have been made to deal with all the suggested minor issues. The clarity/quality of the Figures has been much improved and as too has the English throughout the text. The discussion has been improved and some brief concluding remarks given. There are a few issues that remain to be dealt with, specifically:
The title for the article should be changed as "In silico and in vivo studies of a tumor-penetrating ..."
There are numerous errors in the abstract to be corrected:
line 3, "few little" -> "little"
line 21, "of for" -> "for", line
line 24, "than of" -> "than"
line 26, "can are able to" -> "can"
line 29, "the having" -> "having"
line 30, "that which" -> "which"
On page 11, "statistically significative" should read "statistically significant", and the authors should give the confidence level in brackets, e.g., "... statistically significant (p < 0.001)." Also, the legend to Figure 4 should include the meaning of the asterisks (i.e., the confidence level of the marked differences should be specified here too.)
The legends to Figures 5 and 6 should state whether the error bars shown are standard deviations or standard errors, and the numbers of replicates should be specified.
Once these additional changes have been made, I consider that the revised manuscript can be accepted for publication.
Obj: pharmaceutics-2197144
Dear Pharmaceutics Editor,
We would like to thank you very much for reconsidering our manuscript entitled In silico and in vivo studies of a tumor penetrating and interfering peptide with antitumoral effect on xenograft models of breast cancer by Marin et al., and for your positive feedback on the article.
We apologize for the errors in the abstract but it seems that in the version that we sent by email to the editorial office, the words to be deleted were marked as crossed out words and the words modified were labeled in red. In the version that is available from the web site, all these marks have disappeared and the words have remained, generating some confusion. We have now fixed this definitely. The modifications are labelled in red.
In the new version of the manuscript, the following modifications have been made, according to your remarks:
- The title of the manuscript has been modified
- The statistical significance on figure 4 has been added on the legend of the figure.
- For figures 5 and 6, the information concerning the error bars has been added.
We have added on the web site the file with the supplementary information.
Reviewer 4 Report
Re: Facing in silico results of a tumor penetrating and interfering peptide with antitumoral effect on xenograft models of breast cancer, by Marin et al.
A variety of therapeutic approaches are necessary to target and treat cancers. The authors of this study are investigating the role of several bi-functional peptides as therapeutics. These peptides combine a tumor penetrating peptide, containing a CendR motif for interaction with neuropilin, with an interfering peptide that can reduce the ability for PP2A and SET to interact. In this study the authors use modeling approaches to predict the interaction of these peptides with neuropilin. This work seems to show little difference between the peptides, whether in their normal or cleaved states. More notable is the experimental work using a xenograft system to examine the effectiveness of the peptides.
While there is merit to the results of this paper, the poor grammar and writing style are very distracting – it needs a lot of editing. Many statements are too vague – For example, in the first paragraph of the results, the authors say that the dockQ score is “on average” higher for the cleaved form. Wouldn’t it be more accurate to say that “for 3 of the 4 peptides examined, the dockQ score was higher upon cleavage”? Other vague words here are “little difference”, “slight improvement” “almost no difference”.
The rigor of the data reporting should be improved, as commented on below. Most of the figures in the PDF version that this reviewer is reading are of very poor quality, making it difficult or impossible to verify the data described in the text. The word “Facing” in the title is not clear to this reader – does this mean a “comparison”?
Here are some specific comments:
Lines 52-54: is there a reference for this information?
Line 186: “Similar results (not shown) were obtained for the other systems.” What other systems?
Line 187: both peptides are referred to in this sentence, but only Figure 1B is referenced. Shouldn’t this include Figure 1C?
IP-GG-iRGD cleaved a and b: these names are very awkward; when written, it is often not obvious if the name is being stated, or if ‘cleaved’ is just a verb in the sentence. Consider renaming, or maybe hyphenating the names.
Figure 4: error bars should be included. While the authors state that there was a therapeutic effect starting at J32 (day 32?), this does not appear to be the case for iRGD-IP at day 35. Please be more specific regarding where statistical significance is observed.
Figure 5: a truncated y axis is somewhat deceptive to the casual observer – until the reader carefully examines the y axis, it suggests a difference greater than reality.
Figure 6: this graph should also contain error bars and some indication of statistical analysis.
Figure 7: microscopy images should contain scale bars, not indications of magnification, which will change depending of the size of the figure. The results should contain some indication of frequency at which the stated characteristics were observed and the number of samples (tumors, mice?) that were examined.
Academic Editor Notes
The authors have revised the manuscript and have satisfactorily addressed most of the issues raised by the four reviewers. All modifications have been made to deal with all the suggested minor issues. The clarity/quality of the Figures has been much improved and as too has the English throughout the text. The discussion has been improved and some brief concluding remarks given. There are a few issues that remain to be dealt with, specifically:
The title for the article should be changed as "In silico and in vivo studies of a tumor-penetrating ..."
There are numerous errors in the abstract to be corrected:
line 3, "few little" -> "little"
line 21, "of for" -> "for", line
line 24, "than of" -> "than"
line 26, "can are able to" -> "can"
line 29, "the having" -> "having"
line 30, "that which" -> "which"
On page 11, "statistically significative" should read "statistically significant", and the authors should give the confidence level in brackets, e.g., "... statistically significant (p < 0.001)." Also, the legend to Figure 4 should include the meaning of the asterisks (i.e., the confidence level of the marked differences should be specified here too.)
The legends to Figures 5 and 6 should state whether the error bars shown are standard deviations or standard errors, and the numbers of replicates should be specified.
Once these additional changes have been made, I consider that the revised manuscript can be accepted for publication
Obj: pharmaceutics-2197144
Dear Pharmaceutics Editor,
We would like to thank you very much for reconsidering our manuscript entitled In silico and in vivo studies of a tumor penetrating and interfering peptide with antitumoral effect on xenograft models of breast cancer by Marin et al., and for your positive feedback on the article.
We apologize for the errors in the abstract but it seems that in the version that we sent by email to the editorial office, the words to be deleted were marked as crossed out words and the words modified were labeled in red. In the version that is available from the web site, all these marks have disappeared and the words have remained, generating some confusion. We have now fixed this definitely. The modifications are labelled in red.
In the new version of the manuscript, the following modifications have been made, according to your remarks:
- The title of the manuscript has been modified
- The statistical significance on figure 4 has been added on the legend of the figure.
- For figures 5 and 6, the information concerning the error bars has been added.
We have added on the web site the file with the supplementary information.
Round 2
Reviewer 1 Report
The authors modified the manuscript following the suggestions. Thus, the paper can be accepted for publication.
Reviewer 4 Report
Re: Facing in silico results of a tumor penetrating and interfering peptide with antitumoral effect on xenograft models of breast cancer, by Marin et al.
A variety of therapeutic approaches are necessary to target and treat cancers. The authors of this study are investigating the role of several bi-functional peptides as therapeutics. These peptides combine a tumor penetrating peptide, containing a CendR motif for interaction with neuropilin, with an interfering peptide that can reduce the ability for PP2A and SET to interact. In this study the authors use modeling approaches to predict the interaction of these peptides with neuropilin, followed by a breast cancer xenograft system to analyze the effectiveness of these peptides.
In this re-submission, the authors have improved many aspects of the paper.
Here are a few minor points still outstanding
1) Figure 1C is not labeled on the figure.
2) Figure 2B. Is there supposed to be something between the parentheses on the y-axis? What does E, C, and H stand for? This should be in the figure legend.
3) Figure 4: no error bars for the two bars on the right side of the figure?